# Preconception Diet Interventions in Obese Outbred Mice and the Impact on Female Offspring Metabolic Health and Oocyte Quality

**DOI:** 10.3390/ijms25042236

**Published:** 2024-02-13

**Authors:** Ben Meulders, Waleed F. A. Marei, Inne Xhonneux, Lien Loier, Anouk Smits, Jo L. M. R. Leroy

**Affiliations:** 1Gamete Research Centre, Laboratory of Veterinary Physiology and Biochemistry, Department of Veterinary Sciences, University of Antwerp, 2610 Antwerp, Belgium; ben.meulders@uantwerp.be (B.M.); waleed.marei@uantwerp.be (W.F.A.M.); inne.xhonneux@uantwerp.be (I.X.); lien.loier@uantwerp.be (L.L.); anouk.smits@uza.be (A.S.); 2Faculty of Veterinary Medicine, Department of Theriogenology, Cairo University, Giza 12211, Egypt

**Keywords:** obesity, obesogenic diet, offspring, metabolism, oocyte, preconception care, diet intervention, mitochondria, epigenetics, proteomics

## Abstract

Obese individuals often suffer from metabolic health disorders and reduced oocyte quality. Preconception diet interventions in obese outbred mice restore metabolic health and oocyte quality and mitochondrial ultrastructure. Also, studies in inbred mice have shown that maternal obesity induces metabolic alterations and reduces oocyte quality in offspring (F1). Until now, the effect of maternal high-fat diet on F1 metabolic health and oocyte quality and the potential beneficial effects of preconception dietary interventions have not been studied together in outbred mice. Therefore, we fed female mice a high-fat/high-sugar (HF/HS) diet for 7 weeks and switched them to a control (CONT) or caloric-restriction (CR) diet or maintained them on the HF/HS diet for 4 weeks before mating, resulting in three treatment groups: diet normalization (DN), CR, and HF/HS. In the fourth group, mice were fed CONT diet for 11 weeks (CONT). HF/HS mice were fed an HF/HS diet from conception until weaning, while all other groups were then fed a CONT diet. After weaning, offspring were kept on chow diet and sacrificed at 11 weeks. We observed significantly elevated serum insulin concentrations in female HF/HS offspring and a slightly increased percentage of mitochondrial ultrastructural abnormalities, mitochondrial size, and mitochondrial mean gray intensity in HF/HS F1 oocytes. Also, global DNA methylation was increased and cellular stress-related proteins were downregulated in HF/HS F1 oocytes. Mostly, these alterations were prevented in the DN group, while, in CR, this was only the case for a few parameters. In conclusion, this research has demonstrated for the first time that a maternal high-fat diet in outbred mice has a moderate impact on female F1 metabolic health and oocyte quality and that preconception DN is a better strategy to alleviate this compared to CR.

## 1. Introduction

According to data from the World Health Organization (WHO), the global prevalence of obesity has almost tripled over the past decades [1]. A crucial contributing factor is the increased consumption of a Western-type diet rich in fats and sugar, leading to excess caloric intake [2]. Obese patients suffer from various metabolic complications like reduced insulin sensitivity, hyperlipidemia, and liver dysfunction [3]. Importantly, assisted reproduction clinics report increased rates of infertility and pregnancy complications (e.g., miscarriage or gestational diabetes) in obese women of reproductive age [4,5]. The oocyte plays a crucial role in the pathogenesis of infertility. Studies have demonstrated that, when donor oocytes from healthy mothers were used in obese individuals, pregnancy rates were restored [6]. In obese women, the disturbed metabolic state might be reflected in the follicular fluid (FF), the microenvironment of the growing and maturing oocyte [7]. This can result in nutrient overload in the oocyte, leading to oxidative stress and mitochondrial (MT) dysfunction. For example, feeding outbred Swiss mice a high-fat diet resulted in increased oocyte reactive oxygen species (ROS), increased MT activity, ultrastructural abnormalities and length, and decreased MT roundness [8].

Oocyte MT dysfunction is a key causative link to subfertility in obese women, as mitochondria are essential organelles for oocyte function, not only through the production of ATP [9,10]. Previous research has indicated that oocyte MT dysfunction may alter the epigenome of the oocyte and embryo [11]. Epigenetic patterns are dynamically changing during oogenesis and embryogenesis [12,13]. Also, the oocyte epigenome is vulnerable to environmentally induced alterations during this period [14]. It was shown that diet-induced obesity in mice alters global DNA methylation and histone modifications in oocytes [15,16]. Depending on which genes are affected, this can influence further development. Also, an obese microenvironment can impact the proteome of the oocyte. In vitro maturation of *bovine* oocytes under lipotoxic conditions resulted in specific pro-survival and adaptive proteomic alterations in the oocyte related to unfolded protein response (UPR), MT dysfunction, lipid metabolism, and oxidative stress [17].

Maternal obesity also impacts offspring (F1) health, even when the F1 is fed a control diet. This maternal impact can be mediated during the pre- and periconceptional periods (through the developing oocyte) but also during pregnancy or even lactation [18]. A great amount of research proved that a maternal high-fat diet in inbred C57Bl/6J mice during these periods will negatively affect F1 metabolic health, characterized by, for instance, increased serum triglycerides (TG) [19], insulin [20], and cholesterol [21]. Very few other studies, all in inbred mouse models, have specifically highlighted the importance of the pre- and periconceptional period for the development of noncommunicable diseases in later life [22,23,24,25]. Focusing on F1 oocyte quality, it has, for example, been established in inbred C57Bl/6J mice that a maternal high-fat/high-sugar (HF/HS) diet for 6 weeks before conception results in altered MT ultrastructure [26], altered MT localization, increased MT membrane potential, and decreased MT mass, ATP content, and citrate content [27]. Also, whether the epigenetic alterations in the mother’s oocyte will persist in the offspring is difficult to predict since the epigenome is reprogrammed after fertilization [13]. In one study, it was, however, demonstrated that, when inbred C57Bl/6J mice were fed an HF/HS diet for 6 weeks before conception, global DNA methylation was increased in F1 oocytes [27]. Whether this maternal imprint on F1 oocyte quality also influences the proteomic profile has not been covered before. Furthermore, the intriguing insights described above were all made in inbred mouse models. It is known that inbred mice already display higher rates of MT ultrastructural abnormalities in their oocytes and that outbred mice are more sensitive to HF/HS-diet-induced reduced oocyte quality [8]. For that reason, outbred models were used in this study, as they may be more intricately linked to human physiology.

In assisted reproduction clinics, obese women are often advised to lose weight before conception to increase the odds of pregnancy success by improving metabolic health and fertility [28,29]. However, univocal guidelines are lacking. Therefore, in a study in our lab, the impact of diet normalization (DN) or caloric-restriction (CR) diet for 2, 4, or 6 weeks as preconception care interventions (PCCI) in HF/HS-diet-fed outbred mice on metabolic health and oocyte quality was evaluated. We demonstrated that glucose response and general oocyte quality were significantly improved after 4 weeks PCCI. However, oocyte MT ultrastructure did not fully recover after preconception CR [30,31]. Therefore, these persisting oocyte MT abnormalities may affect embryo and fetal development and may even be transferred through the female germline to the F1 oocyte [26]. Studies in humans have demonstrated that preconception weight loss improves F1 cardiovascular function [32]. Also, in obese mouse models from mixed backgrounds (B6/129/SvEv), it was found that preconception DN prevented F1 pancreatic beta cell dysfunction [33], liver dysfunction [34], and adipose tissue inflammation [35], and, in an inbred obese mouse model, it was found that metabolic function was improved in the offspring after preconception DN [36]. However, no studies have investigated the effects of PCCIs on F1 metabolic health in an outbred model, and how F1 oocytes may benefit from PCCIs in the mother has not been investigated at all. This knowledge is crucial for preserving the next generation’s fertility.

With this research, we aimed to investigate how a maternal obesogenic preconception diet affects offspring metabolic health and oocyte quality in an outbred model and whether possible negative effects can be alleviated by PCCIs. We hypothesized that a maternal HF/HS diet in outbred Swiss mice negatively affects F1 metabolic health and oocyte quality and that preconception DN or CR in the mother can alleviate these negative effects. To test this hypothesis, we fed 5-week-old female Swiss mice an ad libitum HF/HS diet or ad libitum CONT diet for 7 weeks. Then, for the next 4 weeks, the HF/HS-fed mice were split into three groups: HF/HS, DN, and CR. In the HF/HS group, the mice were maintained on an ad libitum HF/HS diet. In contrast, the mice of the DN group were switched to an ad libitum CONT diet, while the CR group was also switched to CONT diet but only 70% of the amount that was consumed in the CONT group (calculated based on daily consumption, paired feeding regimen). CONT mice were maintained on the ad libitum CONT diet. Ultimately, this resulted in 4 treatment groups: HF/HS, DN, CR, and CONT. Then, all mice were mated with CONT-fed Swiss males. After that, HF/HS mice were fed an HF/HS diet from conception until weaning, while all other groups were fed CONT diet from conception onwards. After weaning, offspring were kept on a chow diet and sacrificed at adult age (11 weeks) (Figure 1). We then aimed to evaluate the effect of maternal HF/HS diet on female F1 metabolic health by measuring serum concentrations of insulin, total cholesterol, glucose, TGs, and non-esterified fatty acids (NEFAs). Also, oocyte quality was investigated by assessing MT ultrastructure, global DNA methylation levels, and proteomic alterations. We also aimed to investigate if PCCIs were able to alleviate possible negative effects of maternal HF/HS diet on these metabolism- and oocyte-related parameters.

## 2. Results

### 2.1. Litter Characteristics

At birth, the number of pups in each litter was counted to assess litter size. Also, at weaning, all pups were sexed to determine the female/male ratio. No significant difference was observed between HF/HS and CONT for litter size (*p* = 1) and female/male ratio (*p* = 0.483). Also, none of the other comparisons resulted in significant differences for both outcome parameters (*p* > 0.05) (Table 1).

### 2.2. Offspring Body Weight

At birth (0 weeks), all pups were weighed collectively per litter. At weaning (3 weeks) and adult age (8 weeks and 11 weeks) all female mice were weighed individually. At birth, no significant differences in total litter weight were observed between the groups (*p* > 0.05). At weaning, body weight was significantly increased in HF/HS compared to CONT (*p* < 0.001). CR was able to alleviate this increase in HF/HS (*p* = 0.001) and restore this to the level of CONT (*p* = 1), while DN was not significantly different from HF/HS (*p* = 1). At 8 weeks and 11 weeks, again, no significant differences were observed between the treatment groups (*p* > 0.05) (Figure 2).

### 2.3. Blood Serum Profile

We measured serum insulin concentrations in 11-week-old female F1 mice using a commercial ELISA kit. Compared to CONT, serum insulin concentrations were significantly increased in the HF/HS group (*p* = 0.019). DN and CR were both unable to reduce insulin levels compared to HF/HS (*p* = 0.517 and *p* = 1). However, a numerical decrease in DN compared to HF/HS was observed, making it also not significantly different compared to CONT (*p* = 1). In contrast, insulin concentrations in CR were significantly increased compared to CONT (*p* = 0.026) (Figure 3).

The remaining serum of the female F1 mice was analyzed in a commercial laboratory to determine serum cholesterol, glucose, TG, and NEFA concentrations. Interestingly, no differences were observed between any of the treatment groups in all these parameters (*p* > 0.05) (See Appendix A).

### 2.4. Oocyte Mitochondrial Ultrastructure

#### 2.4.1. Mitochondrial Morphology

Using transmission electron microscopy (TEM), we examined the effect of maternal HF/HS diet and PCCI on F1 oocyte MT ultrastructure. Mitochondria were classified as previously described [8]. Firstly, mitochondria with a spherical homogeneous shape or regular vacuoles were classified as normal. On the other hand, mitochondria were classified as abnormal if they showed a loose inner membrane (spherical or non-spherical), dumbbell shape, elongation, degeneration, rose-petal appearance, or increased electron density (See Appendix A). Compared to CONT, the percentage of MT abnormalities was significantly increased in F1 oocytes from HF/HS mothers (*p* = 0.008). This increase was not restored to the level of CONT in the DN group (*p* = 0.905), while, in CR, even more MT abnormalities were observed in the F1 oocytes compared to HF/HS (*p* < 0.001) (Table 2).

#### 2.4.2. Mitochondrial Dimensions

We also measured MT ultrastructural dimensions in these oocytes using TEM. MT area was significantly increased in F1 oocytes from HF/HS mothers compared to CONT (*p* < 0.001). In DN, the area was decreased compared to HF/HS (*p* = 0.035) and restored to the level of CONT (*p* = 0.598). CR was unable to restore this increase in area in the HF/HS group (*p* = 0.214). MT width was significantly increased in HF/HS compared to CONT (*p* < 0.001). Both DN and CR failed to alleviate the increase that was observed in HF/HS (*p* = 0.059 and *p* = 1.00). MT length was also significantly increased in HF/HS compared to CONT (*p* < 0.001). Both DN (*p* < 0.001) and CR (*p* = 0.013) significantly reduced MT length compared to HF/HS. In both cases, this was reduced to the level of CONT (*p* = 0.279 and *p* = 0.127, respectively). The roundness was not significantly altered in HF/HS compared to CONT (*p* = 0.084). Compared to CONT, MT mean gray intensity was significantly increased in HF/HS oocytes (*p* < 0.001). Compared to HF/HS, DN (*p* < 0.001) and CR (*p* = 0.028) both alleviated this increase. In DN, this was reduced to lower levels than CONT (*p* < 0.001), while, in CR, this was reduced to the level of CONT (*p* = 1.00) (Table 3).

### 2.5. Oocyte Global DNA Methylation

Oocyte global DNA methylation levels were determined using 5-methylcytosine (5mC) immunostaining. We found that 5mC mean gray intensity was increased in HF/HS compared to CONT (*p* = 0.003). Compared to HF/HS, global DNA methylation levels are decreased in DN (*p* < 0.001) but not in CR (*p* = 0.191). Compared to CONT, global DNA methylation levels from both DN (*p* = 0.576) and CR (*p* = 0.842) were not significantly different (Figure 4).

### 2.6. Oocyte Proteome Analysis

Finally, pools of oocytes from all mice of the same litter were analyzed using Ion Mobility Spectrometry–Time-Of-Flight (TIMS-TOF) shotgun proteomic analysis. When all treatment groups were compared, one cluster of 17 differentially regulated proteins (DRPs) was observed, which contained a pattern where these proteins were downregulated in the HF/HS group compared to CONT. The DN group was similar to CONT, while, in CR, some proteins were also downregulated compared to CONT. The pathways in which the downregulated DRPs were enriched are cellular response to hypoxia, cellular response to stress, and negative regulation of intracellular signal transduction (Figure 5). The DRPs included proteasome inhibitor subunit 1 (Psmf1; plays an important role in proteasome function), protein disulfide isomerase associated 3 (Pdia3; a protein folding chaperone), mitochondrial ribosomal protein S36 (Mrps36; helps with protein synthesis within the mitochondrion), ATP synthase inhibitory factor subunit 1 (Atp5if1; involved in mitochondrial depolarization and negative regulation of ATPase activity), myotubularin related protein 14 (Mtmr14; involved in dephosphorylation), cytochrome c oxidase subunit 5a (Cox5a; terminal enzyme of the mitochondrial respiratory chain), and eukaryotic translation initiation factor 2s3x/y (Eif2s3x/y; involved in early steps of protein synthesis) (See Appendix A for full list of DRPs).

## 3. Discussion

In this study, we aimed to investigate if a maternal HF/HS diet in outbred Swiss mice negatively affects F1 metabolic health and oocyte MT ultrastructure, DNA methylation, and proteome and if preconception DN or CR can alleviate these negative effects. Maternal HF/HS diet significantly increased F1 serum insulin concentrations compared to CONT. Also, MT ultrastructural abnormalities, MT size, and MT mean gray intensity were moderately but significantly increased in HF/HS F1 oocytes. Additionally, global DNA methylation levels were increased in HF/HS F1 oocytes, while several DRPs were downregulated. When comparing both PCCIs, preconception DN was capable of reversing most maternal HF/HS-diet-induced effects on F1 metabolic health and oocyte quality. In contrast, this was not always the case when CR was applied as a PCCI.

The effects of the different treatments at the level of the F0 mice were previously published [30,31]. It was shown that, in the HF/HS group, the maternal body weight was significantly increased compared to CONT and that, due to preconception DN and CR, this was restored to the level of CONT after 16 and 8 days of intervention, respectively. Also, serum glucose and insulin concentrations were increased in F0 HF/HS compared to CONT, but, after preconception CR, this was restored to the level of CONT. The same pattern was observed for serum cholesterol but, here, DN and CR both alleviated the HF/HS-induced increase. Focusing on F0 oocyte quality, the percentage of abnormal mitochondria was increased in HF/HS compared to CONT and this was only restored after DN. So, given that in F0 the metabolic health is affected by HF/HS diet and this is only partially prevented by PCCIs, this can alter the microenvironment of the developing F0 oocytes and therefore result in epigenetic alterations. This might ultimately affect F1 development and postnatal health. Also, we assume that the alterations in MT ultrastructure in F0 oocytes can directly impact the development of the offspring and indirectly impact F1 oocyte quality with a possible impact on the next generation. Finally, the uterine environment and milk composition might be different between the treatment groups and, therefore, the fetal and neonatal development of the F1 generation could also be affected. Therefore, we assessed the F1 metabolic health and oocyte quality in this study. We only focused on female offspring to assess how metabolic health and oocyte quality might be associated with each other and to minimize variation due to sex differences.

Firstly, metabolic health in the female offspring was assessed by measuring insulin, total cholesterol, glucose, TGs, and NEFAs in serum, as these were all affected by HF/HS diet in the mothers (except for NEFAs, which were not measured) [30,31]. Due to metabolic programming of the oocyte and/or developing embryo and fetus, these metabolic alterations might persist in the offspring [37]. It has indeed been demonstrated in previous research that these metabolites are altered in the serum of offspring born from HF/HS-fed mothers [19,20,21]. Unexpectedly, we observed that only serum insulin was increased in HF/HS offspring compared to CONT, while the other serum parameters were not altered. The lack of significant differences between treatment groups could be explained by the fact that we used an outbred mouse strain, where genetic variability is higher, while, in the other research, inbred mice were used. Also, in our study, the composition of the HF/HS diet was different and it was fed for 7 weeks before mating compared to 6 weeks in the other studies. Overall, it seems that, except for serum insulin concentrations, the metabolic health defects that are observed in the HF/HS mothers are not transmitted to the offspring and the effects of the PCCIs are also different. It might be believed that there has not been any metabolic programming during oocyte, embryo, or fetal development. However, given that this is a well-described process, it is more likely that feeding the offspring chow diet from weaning (3 weeks) until sample collection (11 weeks) has resulted in a correction of these metabolic alterations. Interestingly, DN was able to alleviate the maternal HF/HS-diet-induced increase in F1 serum insulin concentration and correct this to the level of the CONT group, while this was not the case for CR.

Previous research has established that a maternal high-fat diet in outbred mice results in increased F1 oocyte MT ultrastructural abnormalities compared to control [38]. In our results, we showed that the percentage of abnormal mitochondria was slightly increased in HF/HS oocytes compared to CONT. More specifically, the percentages of degenerated mitochondria and mitochondria with increased electron density were increased. Some would explain this by the transfer of aberrant mitochondria from the mother to the offspring due to an inability of the oocyte to activate mitophagy [8,26,39,40]. However, it might also be that altered metabolic and epigenetic programming in the offspring may result in MT adaptations and persistent metabolic alterations that will eventually lead to de novo MT dysfunction. In the F0 mice, the increase in MT ultrastructural abnormalities in HF/HS compared to CONT was 9%, and this was associated with decreased developmental competence of these oocytes after in vitro fertilization (IVF) [31]. Also, in another study, increased oocyte MT ultrastructural abnormalities were associated with alterations in MT membrane potential (MMP) and reactive oxygen species (ROS) accumulation in oocytes, as well as abnormal lipid droplet accumulation [38]. In contrast, here, in the offspring, the increase in MT ultrastructural abnormalities in HF/HS compared to CONT was only 3.2%. Although this difference was statistically significant, the biological relevance of a 3.2% increase in MT ultrastructural abnormalities is debatable. None of the PCCIs in the mothers were able to alleviate the increased percentage of total MT abnormalities in F1. Even more surprisingly, preconception CR resulted in an increased percentage of MT abnormalities compared to HF/HS. This is in contrast with what was observed in the F0 mice, where the percentage of oocyte MT ultrastructural abnormalities was reduced by preconception DN for 4 weeks and was unchanged in the preconception CR group compared to the HF/HS diet group [31]. What we did observe is that the percentage of mitochondria with increased electron density was reduced in both DN and CR to the level of CONT. Functionally, increased MT electron density is linked with increased glucose utilization and oxygen consumption and is thus indicative of increased oxidative phosphorylation [9]. However, conclusions based on the percentage of mitochondria with increased electron density should be made with caution, as the increase in HF/HS compared to CONT was only 1.3%. The increased percentage of degenerated mitochondria in HF/HS oocytes compared to CONT was also very subtle (1.6% increase) and was not alleviated by the PCCIs. Degeneration of mitochondria can be caused by the lack of machinery in oocytes to remove damaged mitochondria [39]. 

We also observed that, in the HF/HS group, oocyte mitochondria had an increased area, width, length, and mean gray intensity compared to CONT. Again, the differences are subtle but significant. In DN, area and length were restored to the level of CONT, and mean gray intensity was decreased to even lower levels than in CONT. In CR, only length and mean gray intensity were restored to the level of CONT. In previous research, it was shown that a maternal HF/HS diet for 6 weeks in inbred mice resulted in an increase in MT area in F1 oocytes compared to control, which is in line with our results. However, MT roundness was also decreased here in the HF/HS F1 mice, while we did not observe an effect on the MT roundness [26]. The increased MT width, length, and area in the HF/HS oocytes can be indicative of MT swelling, which is caused by the opening of the MT permeability transition pore [41]. As a consequence, the mitochondria lose homeostatic control over their volume. Previous research has demonstrated that a maternal high-fat diet increases MT swelling in the oocyte [42], but this has never been investigated in F1 oocytes. The increased mean gray intensity of the oocyte mitochondria can be linked with the increased electron density that was discussed in the previous paragraph. Importantly, alterations in MT ultrastructure are not always linked with functional defects. If an oocyte contains defective mitochondria, it is capable of compensating for this by several adaptive mechanisms [43,44,45]. This, in combination with the subtle differences, might mean that the observed increases in MT ultrastructural alterations that were demonstrated here might be too low to have a functional impact despite statistical significance. Because of this, other parameters of oocyte quality were also assessed.

By measuring global DNA methylation levels in F1 oocytes, we aimed to gain more insight into how oocyte epigenetic programming might be altered. Interestingly, we observed an increase in global DNA methylation levels in HF/HS oocytes compared to CONT. This is in line with previous studies [27]. We might explain the HF/HS-induced effects on global DNA methylation of F1 oocytes by alterations in imprinted genes of the F0 oocytes. These genes can escape DNA methylation reprogramming after fertilization, meaning that altered DNA methylation patterns in maternal MII oocytes that were induced in the preconception period could be passed to the offspring [46]. When imprinted loci are inappropriately methylated, this can alter gene expression [47], resulting in health effects like neonatal diabetes [48]. Also, global DNA methylation levels increase during folliculogenesis and reach maximal levels at the MII stage [49], which is the stage where we collected the oocytes. An altered follicular microenvironment in the F1 mice linked to the observed increase in serum insulin concentrations in the HF/HS group during this period might disrupt the establishment of epigenetic marks. Another explanation might be that the uterine environment was altered due to the maternal HF/HS diet. As the proliferation and migration of primordial germ cells (PGCs) involve massive DNA demethylation [50], this might also be a sensitive window for environmentally induced alterations. This may also lead to a direct impact on the 5mC levels of PGCs in the developing fetus, which will later develop into oocytes. The increased 5mC in HF/HS F1 oocytes was alleviated to the level of CONT by DN, while, in CR, there was a slight numerical decrease but no significant difference compared to HF/HS.

Finally, the proteome of the F1 oocytes was also analyzed, since most cellular stress responses are expressed at the proteome level [51]. When all treatment groups were compared, a cluster of 17 DRPs was found where the proteins were downregulated in the HF/HS group compared to CONT. The DN group was similar to CONT, while, in CR, some proteins were also downregulated compared to CONT. Interestingly, this pattern where HF/HS is significantly different compared to CONT and DN is more similar to CONT while CR is similar to HF/HS is also what was observed in most of the other outcome parameters. Overall, this suggests that DN is a more promising strategy to alleviate the maternal HF/HS-diet-induced effects on F1 metabolic health and oocyte quality compared to CR. The pathways in which the downregulated DRPs were enriched, are cellular response to hypoxia, cellular response to stress, and negative regulation of intracellular signal transduction. This fits in the well-studied concept of cellular stress responses in oocytes after exposure to a lipotoxic insult [52]. Here, we show for the first time that such proteomic alterations are still present in the oocytes from offspring of HF/HS-fed mothers despite the fact that they were not exposed to HF/HS after weaning and that they can be alleviated by preconception DN in the mothers. It is, however, important to consider that the amount of identified proteins after filtration was 851 and that, of these, only 17 were differentially regulated, making the effects on the proteome subtle. 

An essential element in our experimental design that should be highlighted is the pregnancy and lactation period. Although we aim to investigate the effects of HF/HS diet on F1 oocyte quality and how both PCCIs can alleviate these effects in the preconception period, HF/HS is also fed an HF/HS diet during pregnancy and lactation, while CONT, DN, and CR were exposed to a control diet during these periods. Therefore, differences between HF/HS and all other treatment groups can also be mediated during pregnancy and lactation. The first important notion about this is that we did not aim to investigate windows of sensitivity in this study. This is merely a conceptual study where we tried to gather insights into how maternal diet and PCCIs affect F1 metabolic health and oocyte quality. Also, a significant increase in serum insulin concentration and the percentage of MT abnormalities was observed in CR compared to CONT. This can be mediated through the obese environment earlier than 4 weeks before conception and/or caloric restriction during follicular activation, oocyte growth, and final oocyte maturation. Importantly, DN and CR were significantly different from each other in a lot of outcome parameters: body weight at 3 weeks, percentage of total oocyte MT abnormalities, and oocyte MT width, length, roundness, and mean gray intensity. The only period during which these groups were exposed to a different diet was the preconception period. This stresses the importance of the preconception period in mediating the observed effects on F1 oocyte quality. Further research is required to investigate the importance of these specific windows. Also, further follow-up research is required to investigate the reproductive fitness of the F1 mice as a functional confirmation of the assessed oocyte quality parameters.

It is clear that maternal HF/HS diet in outbred F0 mice only affects F1 serum insulin levels when focusing on metabolic health. On the other hand, F1 oocyte quality is affected at many levels but the differences are often subtle. We know from previous studies that oocyte MT ultrastructural alterations, altered DNA methylation patterns, and proteomic alterations are associated with reduced fertility [9,17,42,53]. Interestingly, preconception DN in the mothers for 4 weeks can alleviate most of the observed negative effects on oocyte quality in her daughters. In contrast, preconception CR in the obese mother seems to be too drastic, as a lot of the detrimental oocyte effects in the HF/HS offspring were not alleviated or even exaggerated after this intervention. This is in line with what was observed in the F0 oocytes [30,31]. Our present conceptual study resulted in interesting findings clearly indicating significant yet subtle effects of preconception maternal health on the daughter’s serum insulin concentration and oocyte quality. Extrapolating these findings to the human setting should be conducted with great caution, as mice are poly-ovulatory animals. However, the outbred Swiss mouse model that was used in this experiment is more relevant to human physiology compared to inbred mouse strains [8]. Furthermore, previous research in our laboratory has established that this strain is sensitive to an obesogenic diet, which is illustrated by metabolic alterations and reduced oocyte quality [8,30,31]. When relating these diets to the human situation, the HF/HS diet resembled a Western-type diet rich in fats and sugar, while the CONT diet corresponded to the daily maintenance requirements, with relatively lower amounts of fats and sugar. In the DN group, the HF/HS diet was switched to a CONT diet. In contrast, the CR mice also switched from HF/HS to CONT diet but were limited to only 70% of the amount that was consumed in the CONT group. This strict dietary regime is aimed towards faster weight loss [54]. An important limitation that follows from this is that normally, when obese women undergo a CR diet, they consult a dietician to ensure sufficient intake of essential micronutrients. In this study, we did not account for this and, therefore, it could be that the CR mice also suffered from a decreased uptake of essential micronutrients, potentially affecting the results. Finally, apart from DN and CR, other PCCIs have been suggested and investigated. For example, it has been demonstrated that the effect of maternal HF/HS diet on oocyte MT morphology can be alleviated by preconception exercise [55]. It would be interesting to assess how this can affect oocyte quality in the offspring in future research.

## 4. Materials and Methods

### 4.1. Animals, Diet, and Experimental Design

The animal procedures were performed in accordance with the relevant guidelines and regulations. Five-week-old female outbred Rj:Orl Swiss mice were fed ad libitum HF/HS (*n* = 18) or CONT (*n* = 6) diet for 7 weeks. The HF/HS diet was composed of 60 kJ% fat (from beef tallow), 20kJ% carbohydrate, and 20kJ% protein (Ssniff (Soest, Germany) E15741-34; corresponding to Research Diets D12492) and 20% (*w*/*v*) fructose (Merck (Hoeilaart, Belgium), 102109450) was added to the drinking water. CONT was a matched purified control diet and this was composed of 10 kJ% fat, 70kJ% carbohydrate, and 20 kJ% protein (Ssniff E157453-04; corresponding to Research Diets D12450J). CONT mice received water ad libitum. After these 7 weeks, HF/HS-fed mice were split into 3 groups, which received a different diet in the next 4 weeks: HF/HS, DN, and CR. In the HF/HS group (*n* = 6), the mice were maintained on an ad libitum HF/HS diet. In contrast, the mice of the DN group (*n* = 6) were switched to an ad libitum CONT diet. Finally, the CR mice (*n* = 6) received the same CONT diet but only 70% of the amount that was consumed in the CONT group (calculated based on daily consumption, paired feeding regimen). CONT mice (*n* = 6) were maintained on the ad libitum CONT diet. Ultimately, this resulted in 4 treatment groups: HF/HS, DN, CR, and CONT. Then, all mice were mated with CONT-fed Rj:Orl Swiss males (*n* = 6) in a crossover design (i.e., each male is used in every group) and maintained on their diets during pregnancy and lactation, except CR who switched to the ad libitum CONT diet. After weaning at 3 weeks, F1 mice were kept on a standard chow diet for 8 weeks (Figure 1). At birth (0 weeks), litter sizes were counted and all mice were weighed collectively per litter. At weaning (3 weeks) and adult age (8 weeks and 11 weeks), all female mice were weighed individually.

### 4.2. Serum Collection and Analysis

At 11 weeks, female mice (*n* = 10/treatment) were fasted overnight and sacrificed by decapitation. Blood was immediately collected and stored at room temperature for 30 min and then at 4 °C until further processing. Then, the blood was centrifuged for 2 min (6000× *g*, 4 °C). After this, serum was collected and divided over two aliquots (20 µL for insulin analysis and 200–350 µL for other parameters) and stored at −80 °C. Firstly, serum insulin concentrations were determined using the Ultrasensitive Mouse Insulin ELISA Kit (90080, CrystalChem, Zaandam, The Netherlands). Then, the remaining serum was analyzed in a commercial laboratory (Algemeen Medisch Labo (AML), Antwerp, Belgium) on an Abbott Architect c16000 (Abbott, Lake Forest, IL, USA) to measure serum total cholesterol, glucose, TG, and NEFA.

### 4.3. Oocyte Collection

Female 11 weeks old F1 mice (*n* = 20/treatment) were injected intraperitoneally with 10 IU PMSG (Pregnant Mate Serum Gonadotropin, Folligon 1000 IU, MSD Intervet, Boxmeer, The Netherlands) 3 days before sample collection. Also, 48 h after PMSG injection and 13–14 h before oocyte collection, all mice were injected intraperitoneally with 10 IU hCG (human chorionic gonadotropin, Pregnyl 5000IE, MSD Intervet) to induce superovulation. At 11 weeks, mice were sacrificed and oviducts were collected from each mouse and immediately stored in a 15 mL tube containing 1 mL L15* medium (L15 + 50 IU/mL Penicillin G sodium salt). Then, this was transferred to a 60 mm petri dish and the ampullae were punctured using a fine needle to collect the cumulus–oocyte complexes (COCs). Next, all COCs were transferred to a droplet of 100 µL L15* + 6 mg/mL polyvinylpyrrolidone (PVP) in a 35 mm petri dish. In another droplet, 0.3 mg/mL hyaluronidase was added to L15* to denude the oocytes. The COCs were transferred to this mixture and left untouched for 1 min. Then, the COCs were pipetted up and down using a 50 µL pipette and stripper tips fitted on EZ-grip (Origio) to ensure denudation of the oocytes. From two mice per litter, 1 COC was left un-denuded for TEM. The denuded oocytes were washed three times in droplets of PBS containing 1 mg/mL PVP. Finally, four oocytes/mouse were fixed in 4% paraformaldehyde for 5mC immunostaining, and the remaining oocytes were snap-frozen in pools for TIMS-TOF shotgun proteomic analysis.

### 4.4. Transmission Electron Microscopy (TEM)

From each F1 mouse, a COC was fixed in 0.1 M sodium-cacodylate-buffered (pH 7.4) 2.5% glutaraldehyde solution for TEM. Then, this was embedded in 2% agarose blocks, followed by washing in 0.1 M sodium cacodylate–7.5% saccharose solution. The blocks were then incubated for 2 h in a 1% OsO4 solution, followed by dehydration through an ethanol gradient. Ultra-thin sections were obtained using EM-bed812 and were subsequently stained with lead citrate. Examination of these sections was performed with the Tecnai G2 Spirit Bio TWIN microscope (Fei, Europe BV, Zaventem, Belgium) at 120 kV. Each oocyte was subjected to the acquisition of 10–15 random images at a magnification of 16,500×. MT morphology was assessed and categorized in a blinded manner according to previously established criteria [8]. Also, width, length, roundness, and mean gray intensity of +/− 100 mitochondria per oocyte were measured using ImageJ software (version 1.53).

### 4.5. 5-Methylcytosine Immunostaining

Oocytes were fixed in 4% PFA (*w*/*v*) for 60 min and stored in PBS-PVP. Immunostaining and imaging were performed as previously described [56,57] using a 1:1600 dilution of rabbit 5mC primary antibody (Cell Signaling, Danvers, MA, USA) and a 1:200 dilution of goat anti-rabbit FITC secondary antibody (Thermo Fisher Scientific, Waltham, MA, USA). For the negative controls, the primary antibody was replaced by an equivalent concentration of normal rabbit IgG. In a glass bottom dish, four droplets of 1 µL L15* were made and this was covered with 2 mL mineral oil. Oocytes were then transferred to the L15* droplets (1 droplet per pup). An SP8 confocal microscope (Leica, Machelen, Belgium) equipped with a white laser source (WLL) at excitation/emission 488/525 nm (to visualize FITC-labelled 5mC) and 530/620 nm (for Texas-red labelled propidium iodide) was used for imaging. The scanned depth was 14 µm with 1 µm interval. Using ImageJ software, the gray-scale intensity of 5mC in the nucleus at each z-stack was quantified and averaged to generate an average mean gray intensity of 5mC for each oocyte (See Appendix A for representative images).

### 4.6. TIMS-TOF Shotgun Proteomics

For each mouse, all collected oocytes (*n* = 10–81 oocytes) were washed in PBS-PVP. Then, they were pooled together with the oocytes of all mice from the same litter (*n* = 64–309 oocytes/pool). Next, these pools were transferred to a 1.5 mL vial in minimal volume, immediately snap-frozen in liquid nitrogen, and stored at −80 °C. Then, these pools were used for TIMS-TOF shotgun proteomic analysis. Using Maxquant, data were filtered, log2-transformed, and annotated based on the mus musculus uniport database. Then, only proteins with valid values in >50% of the matrix were kept and missing data were imputed. For each treatment group, the three samples with the best profiles were used for further analysis in Perseus.

### 4.7. Statistical Analysis

Statistical analyses were performed using IBM Statistics SPSS 29 (for Windows, Chicago, IL, USA). Numerical data were tested for normality of distribution (Kolmogorov–Smirnov test) and homogeneity of variance (Levene’s test), and the means were compared with one-way ANOVA. Litter size, offspring body weight at birth and weaning, and MT area, width, roundness, and mean gray intensity were not normally distributed and thus analyzed using a Kruskal–Wallis test. Categorical data (female/male ratio and MT ultrastructure) were analyzed using binary logistic regression. Post hoc Bonferroni correction was performed sequentially for predefined comparisons: (1) HF/HS vs. CONT, to check whether maternal HF/HS-diet-induced alterations compared to CONT diet; if yes, (2) DN vs. HF/HS and CR vs. HF/HS, to assess if these PCCI’s were capable of alleviating the HF/HS-induced alterations; if yes, (3) DN vs. CONT and CR vs. CONT, to investigate if the PCCIs were able to recover these alterations back to the level of CONT. A *p*-value of ≤0.05 was considered significant (indicated with different superscripts “a” and “b”). Data are expressed as mean ± SEM unless otherwise stated. For the proteomic analysis, FDR 0.1 was used.

## 5. Conclusions

With this research, we demonstrate for the first time that a maternal HF/HS diet in outbred Swiss mice induces subtle but significant effects on F1 metabolic health and oocyte quality. Also, maternal preconception DN seems to be a more promising strategy compared to CR to alleviate these F1 effects.

## Figures and Tables

**Figure 1 ijms-25-02236-f001:**
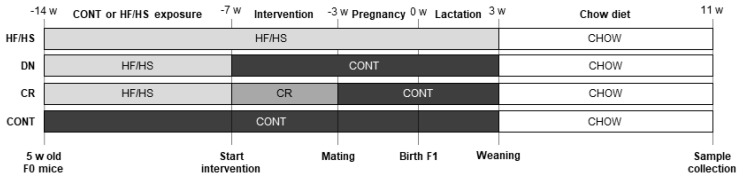
Overview of the experimental design. HF/HS = high-fat/high-sugar; DN = diet normalization; CR = caloric restriction; CONT = control; w = weeks.

**Figure 2 ijms-25-02236-f002:**
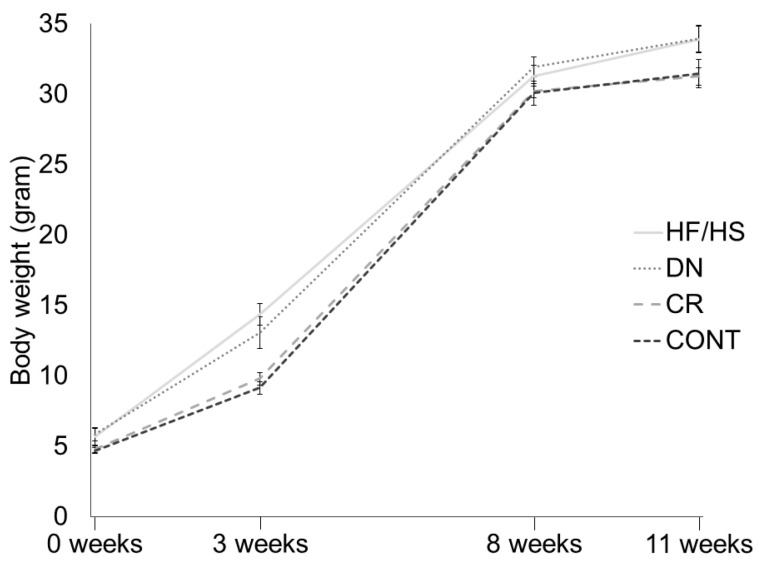
Female offspring body weight at birth (0 weeks), weaning (3 weeks), and adult age (8 weeks and 11 weeks) from offspring born to HF/HS = high-fat/high-sugar diet, DN = diet normalization, CR = caloric restriction, or CONT = control diet. At birth, mice were weighed collectively per litter (*n* = 4 litters/treatment), while, at the other time points, all mice were weighed individually (*n* = 16–20/treatment). Each data point shows mean +/− SEM.

**Figure 3 ijms-25-02236-f003:**
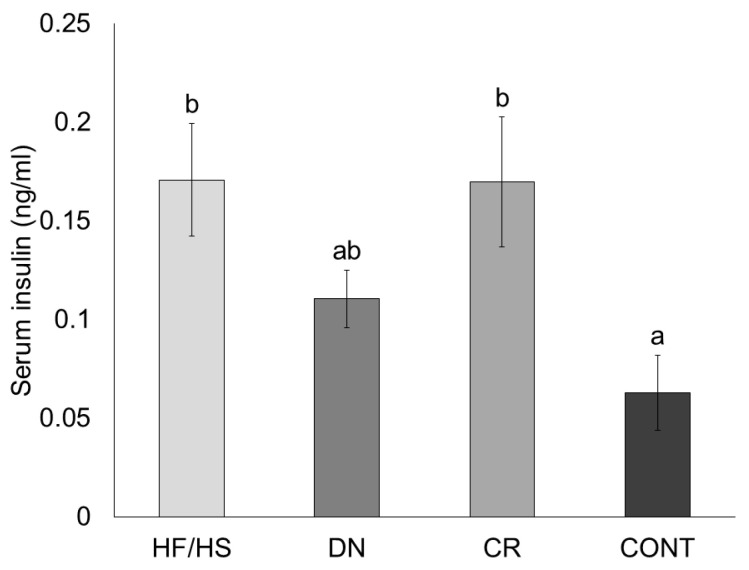
Serum insulin concentrations in female offspring born to HF/HS = high-fat/high-sugar diet, DN = diet normalization, CR = caloric restriction, or CONT = control diet (*n* = 10/treatment). Each bar shows mean +/− SEM. Significant differences (*p* ≤ 0.05) are shown by different letters (a or b).

**Figure 4 ijms-25-02236-f004:**
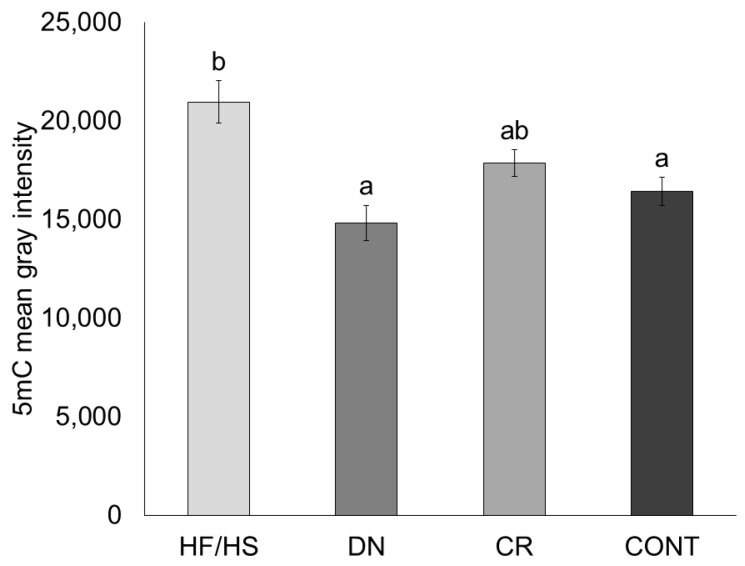
Global DNA methylation levels in oocytes from offspring born to HF/HS = high-fat/high-sugar diet, DN = diet normalization, CR = caloric restriction, or CONT = control diet (*n* = 49–77 oocytes/treatment; *n* = 4 litters/treatment). Each bar shows mean +/− SEM. Significant differences (*p* ≤ 0.05) are shown by different letters (a or b).

**Figure 5 ijms-25-02236-f005:**
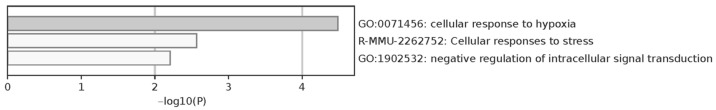
Pathways in which the 17 observed DRPs were enriched (*n* = 3 pools/treatment, *n* = 64–309 oocytes/pool).

**Table 1 ijms-25-02236-t001:** Litter sizes in the different treatment groups (*n* = 4 litters/treatment). HF/HS = high-fat/high-sugar diet, DN = diet normalization, CR = caloric restriction, or CONT = control diet.

	HF/HS	DN	CR	CONT
Litter size (*n*)	12.5 ± 2.1	10.3 ± 1.7	16.5 ± 0.3	15.5 ± 0.9
Female/male (%)	60.3 ± 2.3%	57.8 ± 9.8%	56.7 ± 10.8%	53.4 ± 3.9%

Data are presented as mean ± SEM.

**Table 2 ijms-25-02236-t002:** Morphological classification of mitochondrial ultrastructure in oocytes collected from offspring born to HF/HS = high-fat/high-sugar diet, DN = diet normalization, CR = caloric restriction, or CONT = control diet (*n* = 5–8 oocytes/treatment). IM = inner membrane.

	HF/HS	DN	CR	CONT
Total	1603	1414	774	1172
**Normal**	**1399 (87.3%) ^b^**	**1232 (87.1%) ^b^**	**636 (82.2%) ^a^**	**1061 (90.5%) ^c^**
Spherical homogenous	828 (51.7%) ^a^	830 (58.7%) ^c^	415 (53.6%) ^b^	668 (57.0%) ^c^
Regular vacuoles	572 (35.7%) ^b^	402 (28.4%) ^a^	221 (28.6%) ^a^	393 (33.5%) ^b^
**Abnormal**	**204 (12.7%) ^b^**	**182 (12.9%) ^b^**	**138 (17.8%) ^c^**	**111 (9.5%) ^a^**
Spherical loose IM	79 (4.9%) ^a,b^	77 (5.5%) ^b,c^	58 (7.5%) ^c^	41 (3.5%) ^a^
Non-spherical loose IM	8 (0.5%)	1 (0.1%)	0 (0%)	3 (0.3%)
Dumbbell	13 (0.8%)	8 (0.6%)	6 (0.8%)	5 (0.4%)
Elongation	20 (1.3%) ^a^	26 (1.8%) ^a,b^	15 (1.9%) ^a,b^	33 (2.8%) ^b^
Degeneration	46 (2.9%) ^b^	46 (3.3%) ^b^	44 (5.7%) ^c^	15 (1.3%) ^a^
Rose petal	13 (0.8%) ^a^	19 (1.3%) ^a,b^	14 (1.8%) ^b^	9 (0.8%) ^a^
Increased electron density	28 (1.8%) ^b^	6 (0.4%) ^a^	2 (0.3%) ^a^	6 (0.5%) ^a^

Data are presented as total and proportions (%). Firstly, total mitochondria are shown, followed by total normal mitochondria (with subcategories spherical homogeneous and regular vacuoles), and finally total abnormal mitochondria (with subcategories loose inner membrane (spherical or non-spherical), dumbbell shape, elongation, degeneration, rose-petal appearance, and increased electron density). Significant differences (*p* < 0.05) are shown by different superscripts a, b, or c. From each oocyte, 10–15 random images at a magnification of 16,500× were acquired.

**Table 3 ijms-25-02236-t003:** Oocyte mitochondria ultrastructural dimensions in offspring born to HF/HS = high-fat/high-sugar diet, DN = diet normalization, CR = caloric restriction, or CONT = control diet (*n* = 5–8 oocytes/treatment).

	HF/HS	DN	CR	CONT
Area (pixels^2^)	0.104 ± 0.002 ^b^	0.099 ± 0.002 ^a^	0.098 ± 0.002 ^ab^	0.092 ± 0.001 ^a^
Width (pixels)	0.334 ± 0.003 ^bc^	0.327 ± 0.004 ^ab^	0.341 ± 0.004 ^c^	0.304 ± 0.003 ^a^
Length (pixels)	0.399 ± 0.004 ^c^	0.356 ± 0.004 ^a^	0.381 ± 0.005 ^b^	0.367 ± 0.004 ^ab^
Roundness	0.742 ± 0.004 ^a^	0.729 ± 0.005 ^a^	0.771 ± 0.007 ^b^	0.723 ± 0.005 ^a^
Mean gray intensity	187.2 ± 1.2 ^c^	174.7 ± 1.1 ^a^	184.1 ± 1.1 ^b^	182.1 ± 1.1 ^b^

Area = mitochondrial area; Width = width of the mitochondria; Length = length of the mitochondria; Roundness = 4 × area/(π × (major_axis)^2^); Mean gray intensity = average gray value of the mitochondria. Data are presented as mean ± SEM. Significant differences (*p* < 0.05) are shown by different superscripts a, b, or c. From each oocyte, 10–15 random images at a magnification of 16,500× were acquired.

## Data Availability

In Mendeley Data, all raw datasets are deposited at DOI: 10.17632/6v5mxxvdxh.2. Proteomics data are available via ProteomeXchange via the PRIDE partner repository with the dataset identifier PXD049281.

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
