# Peer review of "Preconception Diet Interventions in Obese Outbred Mice and the Impact on Female Offspring Metabolic Health and Oocyte Quality"

_ijms, 2024, doi:10.3390/ijms25042236_

Round 1

Reviewer 1 Report

Comments and Suggestions for Authors

Major Comments:

The authors need to answer the following queries:

1.   Please explain the novelty of the study in the ‘Introduction’.

2.   Why didn’t the authors mention about the diet regimen for the male mice used for the mating in the study? Please mention this important aspect. If they had not used similar groups of male mice for the mating, please explain the reason? Do they think preconception diet interventions in obese outbred male parent would not impact on female offspring metabolic health and oocyte quality? If so, please explain the reason.

3.   Why didn’t the authors analyse the male offspring metabolic health and sperm quality?

4.   The authors should do the histological study to show the differences in the oocyte qualities among the experimental groups. It would be a very important aspect to firmly comment on the qualities of the oocytes.

5.   Did the authors investigate about the reproductive fitness of the F1 mice?

Minor Comment:

The authors should check the manuscript for the potential grammatical errors.

Comments on the Quality of English Language

The authors should check the manuscript for the potential grammatical errors.

Reviewer 2 Report

Comments and Suggestions for Authors

Although it is an interesting study there some points need improvement

1. Authors do not mention anywhere in their manuscript the aim of their study

2. Materials an methogs usually mentioned after the introduction

5. In their introduction a big part is for a previous study from the same lab. This study was already analyzed,. Authors should omit this part and only mention it in order to explain their scope

Comments on the Quality of English Language

Moderate english editing is required

Reviewer 3 Report

Comments and Suggestions for Authors

In this interesting study Meulders et al. show that offspring of high-fat diet.induced obese mice serum insulin concentrations are significantly elevated and that these offspring have a slightly increased percentage of mitochondrial ultrastructural abnormalities, mitochondrial size, and mitochondrial mean gray intensity in oocytes. DNA methylation was increased, and proteins involved in stress regulation were downregulated in oocytes in offspring of HFD mothers. Strikingly, these alterations were prevented by diet normalization after HFD of others, while calorie restriction after HFD of mothers was less effective or even detrimental. Thus it was suggested that preconception diet normalization may be a better strategy to alleviate obesity-mediated infertility and pregnancy complications compared to caloric restriction.

The experimental design, i.e. the 4 different feeding regimes, is very cleverly conceived and also outbred mice to better reflect a human physiology the inbred are well chosen. Generally, the manuscript is very well written, methods are sound and well described, data well presented and results elaborately discussed.

Minor remarks

I´d suggest to mention the “n”s rather in the figure legend than in the main text to make figures readable independently of the main text.

I really hope that this study promotes further research, particularly in clinical studies. However, extrapolation of results in mice is generally a critical issue. In this case, translation of the experimental setup is also very difficult: a major finding of this study is the superiority of diet normalization to calorie restriction. I´d like to urge the authors to include some discussion, how this could look like for the human (study) situation.

Round 2

Reviewer 1 Report

Comments and Suggestions for Authors

The manuscript is now improved significantly.

Author Response

Dear reviewer,

Thank you for acknowledging that the manuscript has improved significantly.

Kind regards,

Ben Meulders